# Nanomaterials Used in Fluorescence Polarization Based Biosensors

**DOI:** 10.3390/ijms23158625

**Published:** 2022-08-03

**Authors:** Yingqi Zhang, Howyn Tang, Wei Chen, Jin Zhang

**Affiliations:** 1Department of Chemical and Biochemical Engineering, University of Western Ontario, London, ON N6A 5B9, Canada; yzha2268@uwo.ca (Y.Z.); wchen645@uwo.ca (W.C.); 2School of Biomedical Engineering, University of Western Ontario, London, ON N6A 5B9, Canada; htang248@uwo.ca

**Keywords:** nanomaterials, fluorescence polarization, signal amplifier, biosensor

## Abstract

Fluorescence polarization (FP) has been applied in detecting chemicals and biomolecules for early-stage diagnosis, food safety analyses, and environmental monitoring. Compared to organic dyes, inorganic nanomaterials such as quantum dots have special fluorescence properties that can enhance the photostability of FP-based biosensing. In addition, nanomaterials, such as metallic nanoparticles, can be used as signal amplifiers to increase fluorescence polarization. In this review paper, different types of nanomaterials used in in FP-based biosensors have been reviewed. The role of each type of nanomaterial, acting as a fluorescent element and/or the signal amplifier, has been discussed. In addition, the advantages of FP-based biosensing systems have been discussed and compared with other fluorescence-based techniques. The integration of nanomaterials and FP techniques allows biosensors to quickly detect analytes in a sensitive and cost-effective manner and positively impact a variety of different fields including early-stage diagnoses.

## 1. Introduction

Since the term “fluorescence” was introduced by Stokes in 1852, many phenomena and properties of fluorescent materials have been observed and applied in the detection and analysis of chemicals and biomolecules [1,2]. Fluorescence techniques have been employed in various fields such as in biosensors, drug discovery, and bioimaging [3,4,5] Biosensing systems that use different fluorescence mechanisms, including fluorescence polarization, fluorescence resonance energy transfer, fluorescence intensity endpoint measurement, fluorescence-lifetime imaging microscopy and time-resolved fluoroimmunoassay, have demonstrated outstanding performance in terms of quick detection of specific analytes [3,4,5,6,7].

Fluorescence polarization (FP) was theoretically described by Perrin in 1926 [8]. As a sensitive signal-transduction approach, FP has been applied in many fields and in studying the rotational motion, mobility and interaction process of molecules. A fluorophore can emit depolarized fluorescence when it rotates rapidly. In contrast, polarized emission can be observed in fluorophores with slow rotation. Organic dyes are common fluorophores that show different levels of FP depending on molecular weight, solvent viscosity, and molecular rotation [9]. It has been noted that traditional dyes are susceptible to photo bleaching and are very sensitive to environmental factors such as temperature, and pH [10,11]. Figure 1 illustrates the principle of FP. Fluorescence emission from a fluorophore excited by polarized light can be detected in different directions due to the rotating and tumbling of fluorescent molecules in solution, and the degree of polarization of a fluorophore is proportional to its rotational relaxation time (the time when it rotates around 68.5 angle after being excited) [9,12].

The maximum intensity of emitted fluorescent light from the horizontal ($I_{⃫}$) and vertical ($I_{⊥}$) directions are collected to determine the value of FP where polarization (P) can be defined as Equation (1) [13].
(1)$$P=\frac{I_{⊥}-I_{⃫}\;}{I_{⊥}+I_{⃫}}$$

Since FP is also known as fluorescence anisotropy (FA), the value of anisotropy (A) is defined as Equation (2).
(2)$$A=\frac{I_{⊥}-I_{⃫}\;}{I_{⊥}+2I_{⃫}}$$

The relationship between the value of FA and FP can be defined as Equation (3) [13].
(3)$$A=\frac{2P}{3-P}$$

## 2. Conventional FP-Based Biosensing Systems Made of Organic Fluorophores

The principle of FP is used to evaluate the molecular binding reaction in liquid because the fluorescence polarization (FP value) highly depends on the volume, molecular weight, and rotation rate of the fluorescent element. In addition, the rotational rate of organic fluorophores is influenced by temperature and solvent viscosity [14,15]. Figure 2 illustrates the sensing mechanism of a conventional FP-based biosensing system. An organic fluorophore is linked with a receptor. In the presence of an analyte, i.e., targeted molecule, the binding reaction between the analyte and the fluorophore-labelled receptor occurs, which results in an increased FP value because of the increase in size and the slow rotation rate of the fluorescent element after the sensing process, i.e., the formation of targeted molecule-binding the fluorophore-labelled receptor. The differences in the fluorescence polarization before and after the binding reaction is proportional to the concentration of the analyte which normally has a strong affinity with the fluorophore-labelled receptor.

As the value of FP increases with the size/molecular weight of the fluorescent element, it is important to effectively alter the overall size of the fluorescent element during the sensing process. In Figure 3, the FP signal, △mP_1_, after the analyte/targeted molecule reacts with the fluorophore-labelled receptor can be significant enhanced several times to △mP_2_ when a signal amplifier further integrates with the fluorescent element after the sensing process, i.e., targeted molecule-binding the fluorophore-labelled receptor. Therefore, macromolecules, such as proteins, have been involved as a FP signal amplifier to enhance the sensitivity.

Recently, nanomaterials with special fluorescence properties have been studied as a fluorescent label to bind onto the receptor for detecting analytes/targeted molecules by using the FP technique. In addition, compared to other biomolecules used to amplify the fluorescence polarization, nanomaterials that have higher photostability and larger volume/mass show greater potential as signal amplifiers to enhance the performance of sensing systems [16,17]. In this review, different types of nanomaterials used in developing FP-based biosensors have been reviewed. The role of each type of nanomaterial acting as the fluorescent element and/or the signal amplifier has been discussed. Meanwhile, the advantages of FP-based biosensing systems have been discussed and compared with other fluorescence-based techniques.

## 3. Nanostructured FP-Based Biosensors

Nanomaterials applied in FP-based sensing systems can not only be employed to generate fluorescent signals but can also be considered signal amplifiers to enhance the sensitivity of the system by increasing FP.

Nanomaterials with special fluorescence, such as quantum dots (QDs), have the advantages of tunable excitation properties, higher quantum yield and good stability in solution [18,19]. In addition, nanomaterials can be modified with functional groups on their surface due to the high percentage of surface atoms, such as thiol groups on the surface of gold nanoparticles and carboxylic groups on the surface of carbon-based nanomaterials, which result in further bioconjugation of biomolecules/receptors on the surface of nanomaterials [20].

On the other hand, biomolecules with large molecular weights are used to amplify the signal of FP. Unfortunately, most biomolecules are not stable in solvent. Therefore, various nanomaterials have been investigated as amplifiers in mass-based approaches to enhance FP signals to optimize the sensitivity of FP-based assays [21].

Previous publications on using nanomaterials in designing FP-based biosensing systems for generating and amplifying FP signals have been studied. In this review, different nanomaterials are classified into seven categories: carbon-based nanomaterials, metal-based nanomaterials, semiconductor quantum dots, silicon-based nanoparticles, core/shell nanoparticles, inorganic two-dimensional nanomaterials, and polystyrene nanoparticles.

### 3.1. Carbon-Based Nanomaterials

Carbon-based nanomaterials have various formats and structures including graphene oxide, carbon nanotubes, fullerenes, and carbon/graphene quantum dots. The unique thermal, mechanical, electronic and biological properties of carbon-based nanomaterials have garnered much interest [22]. Due to their special properties, these carbon nanomaterials have been extensively studied as biomedical devices for different applications in biosensing, drug delivery, tissue engineering, imaging, diagnostics and cancer therapy [23]. Carbon nanotubes and graphene-related nanosheets with large specific area show excellent performance as amplifiers in FP-based biosensors.

#### 3.1.1. Carbon Nanotubes (CNTs)

CNTs have attracted attention recently for their unique structural and optoelectronic properties, and they have been investigated as effective carriers for biological, imaging-related agents and as a novel water-soluble material.

To detect the inhibitory as well as active effects of DNA methyltransferase (MTase) in solution, Huang et al. developed a multifunctional sensing platform based on multiwalled carbon nanotube (MWCNT) signal amplification and FP. This method used a dye-labeled DNA probe with three segments. Two of the segments were double stranded DNA (dsDNA), one for MTase and the other for recognition by MTase’s corresponding restriction endonuclease, as shown in Figure 4. The last segment of the probe was a piece of single stranded DNA (ssDNA) that could bind to MWCNTs. When there was no MTase in the solution, only a short portion of dye-labeled DNA was cleaved and released by the restriction endonuclease, and since this portion of DNA was not able to bind with MWCNTs, the detected FP value was relatively small. However, when the solution contained MTase, MTase could methylate specific recognition sequences in the dye-labeled DNA probes, protecting them from cleavage by restriction endonucleases. Under strong π-π stacking, the dye-labeled methylated DNA products could bind with MWCNTs to generate new DNA/MWCNTs complexes which significantly increased FP values due to the increased molecular size of the complexes. This assay, using signal amplification of MWCNTs, showed a significant increase in detection sensitivity by two orders of magnitude compared with previously reported assays. This assay can also be applied to the sensitive detection of various DNA MTases and the screening of potential drugs [24].

The MWCNTs-based FP sensor was further developed to detect protease [25]. This new strategy did not use a three-part DNA probe, however. Instead, they used a fluorescein-labeled peptide probe that adsorbed onto MWCNTs via π-π stacking and electrostatic interactions. These fluorescein-labeled peptide-MWCNT complexes presented high FP value due to the increase in molecular weight. When the target protease was present, the protease cleaved the peptide probe specifically, causing peptide fragments carrying fluorescein to split from the complex. As a result, there is a significant decrease in FP value, and the target protease can be detected. In addition, this assay could also theoretically be used to detect protease inhibitors since peptide cleavage, and therefore, the change in FP value, would be inhibited. The detection limits of this assay for thrombin and chymotrypsin were 0.5 pM and 0.3 pM, respectively [25].

#### 3.1.2. Carbon Nanoparticles (CNPs)

Liu et al. reported the development of a signal-amplifying FP sensor (Figure 5) for the detection of apyrase by using carbon nanoparticles (CNPs). This study designed an adenosine triphosphate (ATP)-aptamer (P-ATP) sensor that could adsorb on the surface of CNPs. Due to the increased FP caused by the mass of CNPs and the weak binding capacity of the P-ATP complex to CNPs. Apyrase, an integral membrane protein, could hydrolyze ATP and disassemble the P-ATP complex, which would lead to an increase in FP. The method was highly selective and sensitive for the detection of apyrase. Apyrase concentrations have a linear dynamic range between 0.1 and 0.5 units/μL and a detection limit of 0.05 units/μL [26].

#### 3.1.3. Graphene Oxide

Graphene oxide (GO) is a single layer, two-dimensional nanomaterial that has been used in FP-based biosensors as a signal amplifier to detect a variety of different biomolecules. For example, Qi et al. designed a biosensor that uses GO. This biosensor was developed because there was no significant difference in FP signals between proflavine and proflavine-Rev response element (RRE) complexes. Therefore, proflavine could not be used as a FP probe to identify antagonists of human immunodeficiency virus (HIV) Rev peptide. GO acted as an amplifier to attain FP amplification for the proflavine-RRE-antagonist interaction system [27].

In comparison, Wang et al. also reported a FP platform using GO as an FP signal amplifier that relied on T7 exonuclease-assisted target recycling enhancement. However, while this assay was also used for HIV, it was used to detect HIV DNA, not Rev antagonists. In this sensor, the presence of target DNA led to target recirculation with the help of T7 exonuclease and amplification products were adsorbed onto the GO surface, so that all FP values were enhanced by GO. More importantly, this FP sensor had high detection sensitivity, and was able to detect DNA concentrations as low as 9.12 pmol/L. Under optimal conditions, the change in FP was linearly related to the concentration of target DNA when DNA concentrations were within 0.05 to 0.40 μmol/L [28].

Another example of a GO-based biosensor was that developed by Ye’s research team. This sensor was a GO-based aptamer biosensor that relied on FP to detect aflatoxin B_1_ (AFB_1_). Under π-π stacking and electrostatic interactions, a fluorescein amide (FAM)-labeled aptamer was attached to the surface of GO to construct a novel aptamer/GO macromolecular complex, and thus a larger FP value was detected. However, when AFB_1_ was present, due to its specificity, the aptamer dissociated from the GO surface and bound to AFB_1_, resulting in a change in the molecular weight of the aptamer and therefore a significant change in the FP value. The experimental results showed that when the amount of aptamer was 10 nM, and the concentration of AFB_1_ was within the range of 0.05 to 5 nM, the change in FP and the concentration of AFB_1_ had a good linear relationship. The lowest detection limit was 0.05 nM [29].

On the other hand, carbon quantum dots with tunable fluorescence have the potential to produce FP signals for analysis.

### 3.2. Metallic Nanomaterials

Noble metal-based nanomaterials, including silver nanoparticles (AgNPs) and gold nanoparticles (AuNPs), can be obtained by several different methods, such as chemical reduction, sonochemical, electrochemical and microwave-assisted methods [30]. The shape and size of this class of noble metal nanomaterials can be easily controlled, and different shapes and sizes of noble metal nanomaterials can be gained by changing parameters such as the temperature of solution, the time of reaction and precursors. This controllability makes them good candidates for use as signal amplifiers in various FP-based biosensing systems.

#### 3.2.1. Silver Nanoparticles

Silver nanoparticle overlayers can be used to enhance the fluorescence of adsorbed dye monolayers because of local enhancement of the optical fields near the molecules by interactions with silver plasmons. From the research of Pan et al., deposited AgNPs on a rhodamine red monomolecular layer covalently bound to glass allowed a 20-fold increase in photoluminescence depending on the excitation wavelength and nanoparticle density. At the same time, the fluorescence spectrum was blue shifted and showed a large polarization during emission [31]. For example, Jiang et al. developed a new highly sensitive and selective FP aptamer sensor that detected Hg^2+^ and cysteine using AgNPs signal enhancement technology. The sensor consisted of an aptamer labeled with QDs as a signaler and an aptamer modified with AgNPs as an amplifier. The binding of this new FP aptamer to the target caused a significant difference in the size of the target molecule, resulting in a notable change in the detected FP value. According to the experimental results, the range of detection was 10 nM to 0.4 μM for Hg^2+^ and 20 nM to 0.7 μM for cysteine. The detection limits were 6.6 nM and 11 nM for Hg^2+^ and cysteine, respectively.

Another example is the novel bivalent aptasensor based on FP reported by Chen and his co-workers that was used to detect lactoferrin (Lac) in milk powder (Figure 6). This sensor was found to be more sensitive and accurate than prior methods. They selected two new split aptamers based on previous studies. These bivalent aptamers were modified to bind to fluorescein isothiocyanate (FITC) as a signaling molecule and decahedral silver nanoparticles (Ag_10_NPs) as an enhancer. Since the split aptamers could bind to different parts of Lac, a split aptamer-target complex was formed, resulting in a reduced distance between the Ag_10_NPs and the FITC dye. The results showed that Ag_10_NPs had a metal-enhanced fluorescence (MEF) effect, which also led to an increase in mass. Amplification strategies such as ternary amplification of Ag_10_NPs, splitting of inducers and MEF effect led to a significant increase in the FP values. This analytical method was quite sensitive to Lac detection with approximately three orders higher detection results compared to previously reported assays [32].

Zhu’s research group also developed a biosensing system using Ag_10_NPs to enhance the sensitivity of detection for miRNA-21. In contrast, however, this sensor did not rely on a split aptamer. Instead, this sensor employed Ag_10_NPs linked with two self-assembled complementary ssDNA to be used as a detection probe. It was found that without using an amplifier to detect miRNA-21 at different concentrations, there was no change in FP before and after the recognition of miRNA-21. However, after employing the Ag_10_NPs-ssDNA detection probe, there was an obvious difference in FP. This method showed a wide linear detection range from 100 pM to 16 nM with a limit of detection of around 93.8 pM [33].

#### 3.2.2. Gold Nanoparticles

AuNPs have gained more attention in the field of bioanalysis due to their special optoelectronic properties and excellent biocompatibility [34]. In particular, the strong enhancement effect of AuNPs have caused AuNPs to be widely applied in various biosensing systems to improve detection sensitivity. AuNPs can also easily bind to biological macromolecules such as proteins and DNA without changing their biochemical activities due to the self-assembly principle [35]. Furthermore, recently, AuNPs have been used as labels for amplified detection [36].

Liang et al. developed a fluorescence polarization assay based on AuNPs to detect HIV-DNA. They combined AuNPs with DNA dendritic macromolecules to form AuNP-DNA dendritic macromolecules that act as signal amplifiers. When HIV-DNA was present in the system, the fluorescently labeled DNA probe and AuNP-DNA dendritic macromolecule bound to HIV-DNA and constructed a sandwich-shaped conjugate, resulting in an increase in molecular weight and volume. As a result, the rotation speed of the labeled DNA probe slowed down, causing an increase in fluorescence polarization and system sensitivity. Experimental results showed that HIV-DNA could be detected in concentrations as low as 73 pM, and that when HIV-DNA concentrations was within 150 pM to 6 nM, HIV-DNA concentration and FP had a good linear relationship [37].

Similarly, Wang et al. reported a FP enhancement method based on AuNPs that was used in conjunction with detecting DNA. However, this biosensor did not detect whole strands of DNA, nor did it detect HIV-DNA. Instead, it was used to detect single nucleotide polymorphisms at the nanomolar level. The template strand (containing allele-specific sites) was attached to the surface of AuNPs due to a toehold structure-mediated strand substitution reaction. When fluorescein-labeled blocking DNA was introduced into the solution, they were found to bind to AuNPs via hybridization, which led to a significant increase in FP. However, when the target ssDNA strands in solution separated from the fluorescein-labeled DNA on AuNPs due to substitution reactions, there was a significant decrease in FP (as shown in Figure 7). The experimental results showed that the method developed by Wang et al. could still observe more than 10-fold signal difference in mixed solutions containing 100-fold single-base mismatched strands, and the assay was simple, cost-effective, and widely applicable [38].

Beyond detecting DNA or changes in DNA, AuNPs have also been used to detect other molecules. For example, He and colleagues reported a method with high sensitivity and selectivity to detect ATP based on the enhancement of anti-digoxin (mouse anti-digoxin monoclonal)-AuNPs. ATP that was present in the solution would bind to QD-labeled DNA and antibody-AuNPs, forming DNA hybrids of increased molecular weight and volume, resulting in slower rotation and therefore significantly increased FP values. The sensor could detect ATP at a minimum of 1.8 pM. In addition, in ATP concentrations of 8 × 10^−12^ M to 2.40 × 10^−4^ M, the detection concentrations of ATP showed a good linear relationship with FP. This assay was found to be more sensitive than many other assays by two to three orders of magnitude, and it possessed a wide dynamic range of more than ten orders of magnitude. According to He et al., this approach could be extended to detect different oligonucleotide-dependent analytes [39].

In addition, AuNPs have also been incorporated into sensors that detect specific ions. An example would be the novel biosensor developed by Ye and his colleagues to detect Hg^2+^ that was both sensitive and selective. This method relied on the principle of thymine–Hg^2+^–thymine (T–Hg^2+^–T) coordination chemistry and FP enhanced by AuNPs. Their experimental results showed that this assay had a low detection limit of 1.0 nM, which was about two to three orders of magnitude higher than other assays. In addition, the assay was highly selective and could accurately detect the Hg^2+^ concentration despite the existence of various different metal ions causing interference in samples, and the detection time was reduced to 10 min [40]. Wang et al. also recently designed a simple, cost-effective FP biosensor that used AuNPs to detect ions. However, this biosensor did not detect Hg^2+^ ions, instead it detected Ag^+^ ions. This biosensor relied on cytosine-Ag^+^-cytosine (C-Ag^+^-C) coordination chemistry which involved the specific interactions between Ag^+^ and a C-C mismatch in DNA that led to the formation of a stable Ag^+^-DNA complex. This design involved two complementary DNA probes with C-C mismatches, probe A and probe B, respectively. Probe A was modified onto the surface of a AuNP while probe B was labeled with the fluorophore, fluorescein. In the presence of Ag^+^ ions, probe A and probe B came together through a C-Ag^+^-C interaction. Since the complex had a larger molecular weight, the FP value increased. This method was found to be highly selective, rapid, and sensitive with a linear correlation between the change in FP and Ag^+^ concentrations at 50 to 750 nM, and a detection limit of 9.5 nM. In addition, it was theorized that this biosensor could be applied in detecting other metal ions by changing the DNA sequences of the probes [41].

AuNPs can also be used as carriers to modify the aptamer and increase the molecular weight of analytes to compensate for its lack of polarized exciting light effects on the emitted light. Based on this principle, Samokhvalov et al. used AuNPs with an average diameter of 8.7 nm as an anchoring module with ochratoxin A (OTA) as the target analyte and detected OTA in controlled spiked white wine. The experimental results showed a 25-fold reduction in the detection limit of 2.3 μg/kg compared to the native aptamer [42].

### 3.3. Semiconductor Quantum Dots (QDs)

Nanomaterial with strong fluorescence, such as CdTe quantum dots, have been used in building many types of fluorescent biosensors [43]. For example, CdTe QDs have been implemented into FP-based biosensors to generate FP signals. Their outstanding performance has caused them to be considered as promising alternatives for traditional organic dyes.

Tian et al. reported an aptamer sensor based on CdTe/CdS QDs and mass-amplified FP analysis for the detection of ATP. The ATP aptamer was modified by digoxin antigen and could bind to complementary DNA hybrids modified by CdTe/CdS QDs. Due to the addition of digoxin antibodies, an immunoreaction occurred, resulting in an increase in the mass of the aptamer probe. When ATP was present in the solution, the ATP aptamer recognized ATP and thus separated from the digoxin antibody, leading to a decrease in the polarizability of the fluorescence. In ATP concentrations of 10 to 350 μM, FP and ATP concentration had a linear relationship. In addition, the sensor had a detection limit of 3.7 μM. The assay was also highly selective because even when large amounts of ATP analogues, such as amino acids, proteins, and glucose, were present in the solution, there was no significant change in the results [44].

Similar to Tian et al., Zhang et al. also developed a sensing platform by using CdTe/CdS QDs. However, the purpose of this multifunctional FP sensing platform was for the detection of Hg^2+^, biothiolated amino acids or peptides in samples. The sensing platform used a CdTe/CdS QDs fluorescence polarization probe with a K^+^-mediated G-quadruplex (GQ-DNA) as a signal amplifier. When Hg^2+^ was present in the solution, GQ-DNA and QDs bound to T-Hg^2+^-T due to their strong affinity, forming a complex that increased the molecular size of QDs, leading to a significant increase in FP value as shown in Figure 8. After adding biothiol amino acids to the above solution, they reacted with Hg^2+^ to form thiol-Hg^2+^, resulting in the inability of T-Hg^2+^-T to form and the inability of GQ-DNA and QDs to bind to it to form a large molecular complex, leading to a significant decrease in FP value. The detection limits of this assay were 8.6 nM and 9.9 nM for Hg^2+^ and cysteine, respectively [45].

In contrast to the FP-based biosensor made of QDs, multicolor QDs have been developed for a FP immunoassay that could detect two tumor markers, carcinoembryonic antigen (CEA) and alpha-fetoprotein (α-AFP), simultaneously in human serum. They used QDs 520 and QDs 620 as fluorescent sensors, labeled with α-AFP and CEA, respectively, for FP immunoassay. The results showed that the multi-analytical immunosensor could detect the two tumor markers at minimum concentrations of 0.28 ng/ mL and 0.36 ng/ mL for α-AFP and CEA, respectively. Furthermore, the sensor was able to detect the markers at concentrations of 0.5 ng/mL to 500 ng/mL [46].

Meanwhile, Zhang and colleagues reported a FP platform for detecting influenza A (H1N1). The platform consisted of two parts: a bifunctional protein-binding aptamer (Apt-DNA) and a DNA-functional QDs FP probe. They combined biotin-labeled DNA with QDs as a FP probe and the macromolecular protein streptavidin as an amplifier. In the presence of the target H1N1, the QDs probe showed a significant increase in FP values in the sandwich assay. The assay’s detection limit was 3.45 nM. It was found that FP and H1N1 concentrations had a linear relationship with 10 nM to 100 nM of H1N1 [47].

Another biosensor that used QDs was the sensor developed by Meng’s research group. This sensor was CdTe QDs-based, and it was used to identify hepatitis B virus (HBV) surface antigen’s dominant B-cell epitopes and to detect peptide antigenicity using FP technology. The principle of the technique they applied was that when the QDs-labeled peptide was bound to its corresponding antibody, the rotation of the fluorescent tracer would slow down, the polarization of the emitted light would be enhanced, and the detected FP value would increase. This assay has the advantage of being simple to perform, being less time consuming and being highly sensitive [48].

In contrast, Zhou’s publication used CdTe QDs to determine Hg^2+^ via observing the differences of FP before and after reacting with Hg^2+^. Their results indicated that the reaction between Hg^2+^ and CdTe QDs caused QDs to become unstable which resulted in aggregation of QDs, slowing its rotation in solution, and therefore increasing the FP value. This finding provided a new perspective on the FP technique to explore the mechanism of certain reactions [49].

### 3.4. Silica Nanoparticles

Silica nanoparticles (SiNPs) are easy to fabricate and have shown good biocompatibility. Due to their special properties, such as high porosity and unique surface characteristics, they can immobilize more biological macromolecules, while also providing a larger space for drug delivery [50]. Based on these properties, SiNPs have also been considered as promising amplifiers for the design of FP-based biosensing systems and exhibited outstanding performance.

Chen and colleagues developed a novel method for detecting HBV DNA that applied the principle that biotin-DNA probes could strongly bind to the surface of streptavidin-modified silica NPs (SA-SiNPs) due to the high affinity attraction between biotin and streptavidin. They found that when no fluorophore was present in the system, the FP signal could not be detected. When Ag-DNA probes were added to this system, an intercalation complex of DNA/Ag nanoclusters was formed. The complex was small, however, so relatively small FP values were detected. However, when target HBV DNA was present in the system, it bound to the surface of SA-SiNPs, forming an intercalated structure that increased the molecular size of the SA-SiNPs, resulting in a significant increase in the detected FP values. However, when there was no HBV DNA present in the system, the sandwich structure could not be formed, and the detected FP values were smaller. Therefore, this assay could easily be used to accurately detect the presence of HBV DNA [51].

Another group, Zhao et al. also developed a platform that used SA-SiNPs to detect DNA (as shown in Figure 9). This group investigated and invented a bi-amplified FP detection platform based on hybridization chain reaction (HCR) and streptavidin-modified SiNPs (SA-SiNPs) as signal amplifiers. Two probes labeled by biotin and fluorescein (named H1 and H2, respectively) were used. In the presence of the target nucleic acid, probes H1 and H2 were crossed open autonomously by HCR to form long-notched dsDNA. Due to the formation of long strands, the rotation of fluorescein slowed down and the FP value increased. After adding SA-SiNPs to the above system, the FP value increased significantly due to the specific interaction between biotin and streptavidin, resulting in the binding of the product dsDNA to SA-SiNPs and the formation of macromolecular complexes. Zhao et al.’s FP platform was found to have a detection limit of 35 pM. In addition, the value of FP had a linear relationship with nucleic acid concentrations, but only when the concentrations were within 0 to 2.5 nM [52].

Liao et al. also developed a FP method based on SA-SiNPs. However, in contrast with the two examples above, this FP platform was not used to detect DNA. Instead, this method was used to detect Cu^2+^ and it was based on a combined signal enhancement strategy of click chemistry and SA-SiNPs. They used two DNA strands, named P1 and P2, with one end attached to the alkyne group and the other end attached to the azide group. When Cu^2+^ was present in the system, Cu^2+^ was produced to Cu^+^ by the reduction reaction of sodium ascorbate in the system, and due to the presence of Cu^+^, the system undergoes an azide cyclization reaction and P1 bound to P2. When the biotin marker P3 was added to the above system, P3 could hybridize with P1, leading to the formation of P1P2P3 complexes in the system, and these complexes could attach to the surface of SA-SiNPs, leading to an increase in FP. When there was no Cu^2+^ in the system, P1 and P2 could not combine to form P1P2P3 complexes, and therefore could not be immobilized on the surface of SA-SiNPs, leading to a smaller FP value. The lowest Cu^2+^ concentration detected by this method was 0.0178 μM [53].

Not all FP biosensors that use SiNPs use SA-SiNPs, however. For example, Yue et al. designed a new aptamer biosensor to detect thrombin in human serum that was based on the FP method of SiNPs signal amplification effect. SiNPs were used as an enhancement probe, and streptavidin and biotin were attached to the surface of SiNPs because of specific binding interactions to form thrombin aptamers. When thrombin was present in the system, the thrombin aptamer was induced to form a tetrameric structure. Another thrombin aptamer labeled with luciferin that could bind to the surface of SiNPs to form a sandwich structure complex was added to the above system. Due to the increase in the volume of the complex, the FP value increased significantly. It was found that FP increased linearly in relation to thrombin concentrations at thrombin concentrations of 0.6 to 100 nM, and that the sensor’s detection limit was 0.20 nM [54].

Another example of SiNPs in biosensors is the FP-based approach designed by Huang’s team to detect ATP based on aptamer structure-switching-triggered nanoparticles (NPs) enhancement to amplify FP signals by using SiNPs as amplifiers. Control experiments were obtained by testing the FP value with and without using SiNPs to amplify FP signals. The results indicated that the FP value increased by around 48-fold by applying SiNPs as amplifiers due to SiNPs ability to significantly enlarge the molecular volume of the complexes [55].

### 3.5. Core/shell Nanoparticles

Core-shell nanoparticles consist of two parts, an internal core made of one material, and an external shell made with another material. The overall stability and dispersion of the core/shell nanoparticles can be improved by changing the surface properties. Core/shell nanoparticles with fluorescence are also used to develop FP-based applications to generate signals [56].

#### 3.5.1. Near-Infrared (NIR) Fluorescent Nanoparticles

NIR light, defined as light with wavelengths above 650 nm, has greatly expanded the ability of fluorescence for biomedical applications due to its low absorption and relatively low autofluorescence [57,58]. NIR fluorescent nanoparticles with different core contents and surface ligands have been found to have special properties, leading to applications in bioimaging, drug delivery, and biosensing [59].

Deng’s group synthesized a core/shell NIR fluorescent nanoparticle based on the FP biosensing system for the detection of α-AFP in whole blood samples. The assay applied the principle that NIR fluorescent nanoparticles could bind to polyclonal anti-α-AFP antibodies by conjugation. When α-AFP was present in the system, the α-AFP antigen and its antibody bound together due to specific interactions, forming a complex due to the immunoagglutination phenomenon. Due to the increase in molecular weight, there was an increase in the FP value. The results showed a linear correlation between FP and the detected α-AFP concentrations in the range of 1.9 to 51.9 ng/ml [60].

This research group also reported a rapid homogeneous assay of whole blood target leukemia cells based on FP measurements that relied on the specificity of the DNA aptamer and the signal enhancement by NIR fluorescent nanoparticles. When there were no target cancer cells in the whole blood sample, the size of the aptamer modified NIR fluorescent nanoparticles was small, and the detected FP was also relatively small. When the target cancer cells were present in the whole blood sample, the aptamer bound to its surface membrane protein and formed a complex due to the agglutination phenomenon, resulting in an increase in molecular size and a significant increase in FP value. The results showed that the assay was efficient and rapid, the samples did not need to undergo complex separation steps, and the detected target cancer cells were linear in the concentration range of 4.0 × 10^3^ to 7.0 × 10^5^ cells/mL [61].

In another of the same group’s publications, they reported a sensing platform for the detection of single nucleotide polymorphisms by assembly effects and ligase reactions based on FP detection of core-shell fluorescent nanoparticles. The detection process was divided into two steps. In the first step, two probes (named P1 and P2, respectively) labeled by different nanoparticles bound to the target DNA strand by a hybridization reaction in which perfectly matched probes were joined together in the same solution due to a ligase reaction and no joining occurred between mismatched probes. In the second step, the above solution was warmed up by heat treatment to test the ligation between probes. The formed double chains were denatured by heating, and in the solution with perfectly matched probes, the FP value did not return to the initial value. Meanwhile in the solution with mismatched probes, the assembled fluorescent nanoparticles dispersed, and the FP dropped to the initial values. The results showed that this assay was more sensitive, specific, rapid and convenient than other assays [62].

#### 3.5.2. Core/Shell Semiconductor Quantum Dots

The efficiency of core/shell semiconductor nanoparticles can be improved, i.e., increased quantum yields and reduced reaction times, by selectively cladding the core material with higher bandgap shell materials. For QDs combinations (CdS/HgS/CdS, CdS/CdSe/CdS, etc.), since electrons and holes are confined within the first shell layer with a low band gap, these materials have excellent photoluminescence properties [56].

Tian’s research group reported an immunoassay to detect α-AFP by using the CdTe/CdS/ZnS QDs. Compared with using other methods to determine α-AFP, such as enzyme-linked immunosorbent assays and enzyme-linked fluorescence assays, FP-based detection could avoid the hook effect and background fluorescence of serum components. Compared with FITC, CdTe QDs, CdTe/CdS QDs, core–shell–shell QDs, such as CdTe/CdS/ZnS capped by thioglycolic acid, have the lowest *p* value, are stable, have higher quantum yield and longer fluorescence lifetime. As such, they could be considered the best choice for sensing large molecules. These modified QDs linked with a specific antigen to measure α-AFP showed an ability to detect α-AFP in a range of 1–500 ng/mL [63].

### 3.6. Inorganic Two-Dimensional (2D) Nanomaterials

Two-dimensional (2D) nanomaterials are sheet-like structures that are typically larger than 100 nm and less than 5 nm thick. They are classified into three main categories, one of which is inorganic 2D nanomaterials which will be covered here as they have been used to develop biosensors based on FP. 2D nanomaterials usually have anisotropic chemical and physical properties, good mechanical properties, unique optical properties, and flexible bioactivity. In addition, 2D nanomaterials have a high surface-to-volume ratio, providing a larger surface area with more surface anchoring sites for biomolecules than one-dimensional (1D) and three-dimensional (3D) nanomaterials. Due to these properties, 2D nanomaterials have been used in a variety of applications such as regenerative medicine, energy storage, additive manufacturing, cancer therapeutics, and biosensing [64,65].

#### 3.6.1. MnO_2_ Nanosheets

In recent decades, MnO_2_ nanosheets have been used in a variety of fields including fluorescence sensing, and biomedicine. They are made of a MnO_6_ octahedral crystal lattice with shared edges. Each octahedral is composed of a Mn^2+^ core atom with six oxygen ions, and each oxygen ion is coordinated with three Mn^2+^ [66]. In addition, they are excellent adsorbers, especially for ssDNA, which allows them to act as FP enhancers [67].

For example, Qi et al. designed a sensor (Figure 10) for Ag^+^ that depends on C-Ag^+^-C and guanine-Ag^+^-guanine (G-Ag^+^-G) pairing. This sensor was composed of a proflavine-dsDNA complex that, in the presence of Ag^+^, would bind to Ag^+^ and form C-Ag^+^-C and G-Ag^+^-G pairs, forming proflavine-dsDNA-Ag^+^ complexes. Through electrostatic interactions between the surface of MnO_2_ nanosheets and Ag^+^, the complexes were adsorbed onto the nanosheets. Due to the increase in molecular weight, FP increased. This strategy was found to be highly selective for Ag^+^, as Ag^+^ was the only metal ion capable of inducing a notable change in FP when compared to metal ions such as Li^+^, Mn^2+^, Ca^2+^, Mg^2+^, and Zn^2+^. It was found that this method had a linear response between the change in FP and Ag^+^ concentration in the range of 30 to 240 nM, and it had a limit of detection of around 9.1 nM [67].

Qin and colleagues also developed a biosensor that relied on MnO_2_ nanosheets forming stable complexes with fluorescein-labeled ssDNA. In contrast with the sensor by Qi et al., this biosensor detected organophosphorus pesticides (OPs), which are used in agriculture for increased crop yield and are the cause of food and environmental pollution. When OP was not present, acetylcholinesterase hydrolyzed acetylthiocholine into thiocholine. Thiocholine, in turn, degraded MnO_2_ nanosheets into Mn^2+^ ions, rendering the nanosheets unable to form a complex with fluorescein-labeled ssDNA. As a result, FP was quite low. However, when OP was present, acetylcholinesterase was inhibited, and thiocholine did not form. As a result, the fluorescein-labeled ssDNA was able to adsorb onto the MnO_2_ nanosheets and there was a significant increase in FP. This strategy showed a good linear relationship between the change in FP and the logarithm of OP concentration in the range of 0.01 to 20 ng/mL, and it had a detection limit of 0.01 ng/mL. While Qin et al. used diazinon as a model analyte for OPs, they also tested other common OPs such as omethoate, paraoxon-methyl, chlorpyrifos, and dichlorvos, and the biosensor was found to be sensitive towards all of the OPs. This biosensor was also found to be selective for OPs as there was a negligible change in FP for other pesticides such as acetamiprid, endosulfan and polyvinyl chloride. Furthermore, this sensor could also work in water samples, and had results comparable to a high-performance liquid chromatography (HPLC) assay. As such, this biosensor is an effective, sensitive, and selective method to detect OPs [68].

#### 3.6.2. TiS_2_ Nanosheets

As 2D transition metal dichalcogenide nanosheets, TiS_2_ nanosheets have received much attention because of their electrochemical, electronic, and optical properties. They have been applied in a variety of electronics, and biosensors. In particular, TiS_2_ nanosheets are capable of discerning the differences between dsDNA and ssDNA, which further improves their potential as biosensors.

For example, Li et al. created a biosensor using TiS_2_ nanosheets. It relied on the terminal protection of ssDNA phenomenon where biomolecules bound on the terminus of ssDNA allows that piece of ssDNA to avoid hydrolysis by exonucleases. Two separate ssDNA strands have been designed as shown in Figure 11. The first, probe 1, was an enzyme strand linked to folate at its 3′ end. The second, probe 2, was a carboxy fluorescein-labeled substrate strand that binds to TiS_2_ nanosheets via van der Waals interactions between the nucleotide bases and the basal planes of the nanosheets. When the folate receptor (FR) was not present in the system, probe 1 was hydrolyzed by exonuclease I (Exo I) and it could no longer hybridize with probe 2 into a Zn^2+^-dependent self-hydrolyzing deoxyribozymes (DNAzymes). However, when FR was present, FR would bind to the folate on probe 1, preventing the probe from being digested by Exo 1. As a result, probe 1 and probe 2 can hybridize and form a DNAzyme. Once Zn^2+^ was added to the system, the DNAzyme was activated and probe 2 was cleaved into two short DNA fragments. Since TiS_2_ nanosheets had stronger binding affinity to longer ssDNA strands than shorter strands, the short fluorescein-labeled ssDNA fragment could not adsorb onto the TiS_2_ nanosheet surface, resulting in low FP. Once probe 2 was cleaved by the DNAzyme, the released probe 1 would bind to another probe 2 within the system and the process would repeat, causing the overall FP value to decrease. This sensor had a limit of detection of 0.003 ng/mL and a semi-logarithmic relationship between FR concentrations and the change in FP in the range of 0.01 to 20 ng/mL. When this method was tested on other biomolecules such as human serum albumin, thrombin, IgG, and streptavidin, the resulting change in FP values were significantly lower than the results for FR. Hence, this method was highly selective for detecting FR. In addition, when compared to commercial FR 1 Enzyme-Linked Immunosorbent Assay (ELISA) kits, the results were comparable. Therefore, this biosensor was an effective, and selective strategy for detecting FR [69].

Furthermore, this method could be modified to detect thrombin by replacing the folate on probe 1 with an aptamer DNA sequence that binds to thrombin. It was found that the modified strategy had a good linear relationship between the logarithmic concentration of thrombin and the change in FP over the range of 0.05 pM to 100 nM, with a detection limit of 0.01 pM. In addition, the modified sensor only had a substantial change in FP for thrombin when compared to other human blood biomolecules. Therefore, the biosensor developed by Li et al. was effective, selective, and could easily be modified to detect other biomolecules.

#### 3.6.3. WS_2_ Nanosheets

Other two-dimensional transition-metal dichalcogenides, i.e., WS_2_ nanosheets, have excellent thermal, mechanical, and electronic properties which makes them ideal for use in applications such as biosensing. WS_2_ nanosheets, in particular, are tens to thousands of nanometers long and as such carry large mass, which is ideal for amplifying FP.

Zhao’s publication demonstrated an approach to determining DNA glycosylase activity by applying tungsten disulfide (WS_2_) nanosheets as amplifiers to build a FP-based bioassay. In the presence of DNA glycosylase, many fluorescein fluorophores were not able to be adsorbed on the surface of WS_2_ nanosheets, resulting in a low FP signal. When the system did not have DNA glycosylases, the FAM dye-labeled ssDNA (DP) was bonded to the surface of WS_2_ nanosheets which led to a large FP value. The uracil-DNA glycosylase and the human 8-oxoguanine DNA glycosylase were used to test the effectiveness of this assay and both of them showed better performance than most related studies with a limit of detection of around 0.00030 U/mL and 0.0070 U/mL, respectively [70].

### 3.7. Polystyrene Nanoparticle (PS NPs)

Polystyrene nanoparticles, with their good biocompatibility, have been extensively used in a variety of applications including biosensors. PS NPs with different sizes were prepared to be used in FP-based assays as amplifiers to enhance FP signals. Meanwhile, surface-modified PS NPs can obtain stable colloids in biological fluids with low polydispersity index [71].

Huang’s research group investigated a FP-based biosensing system based on allostery-triggered cascade strand-displacement amplification (CSDA) to detect proteins (as shown in Figure 12). PS NPs were applied as amplifiers in this biosensing system. Many probes were assembled onto PS NPs during the CSDA process, resulting in a significant FP increase, which exhibited a readout signal for detecting proteins. Thrombin (Tb) and vascular endothelial growth factor-165 (VEGF165) were involved as model analytes to test this method’s effectiveness. The limitation of detection for Tb and VEGF165 is about 6 orders of magnitude lower than other sensors [72].

While Li’s study also described a FP-based approach using PS NPs as amplifiers, this sensor did not rely on CSDA nor did it detect proteins. Instead, this sensor was used to determine miRNA based on T7 exonuclease-assisted dual-cycle signal amplification. Since there was an obvious difference in FP values between free carboxyfluorescein and PS NPs-captured intact signal probe after adding miRNA, this method can detect the target miRNA with a limitation around 0.001 nM. In addition, this FP-based biosensor was also capable of measuring target miRNA in total miRNA extracts with high selectivity [73].

## 4. Discussion

Fluorescent techniques, such as fluorescence polarization and fluorescence resonance energy transfer, have been applied in biosensors. Table 1 shows the advantages and disadvantages of different fluorescent techniques in biosensors. Fluorescence intensity (FI) endpoint measurement is an approach that measures the change of fluorescence intensity caused by biological processes [74]. It is a simple process to verify the presence of targeted molecules. However, its signal-to-noise ratio is low. Fluorescence resonance energy transfer (FRET) is a distance-dependent non-radioactive energy transfer process. FRET-based biosensing systems normally analyze the difference of FRET intensity/wavelength in two competitive reactions, which has high signal-to-noise ratio, and can be used to detect samples in both of liquids and solids [75]. Time-resolved fluoroimmunoassay (TRFIA) is a technique developed based on measuring the emission from specific binding of lanthanide ions-conjugated antibody-antigen molecules similar with other fluoroimmunoassays, but lanthanide ions are damaging to environmental and human health, expensive and difficult to obtain [76]. Fluorescence lifetime imaging microscopy (FLIM) is a kind of visible method based on observing the fluorescence decay of fluorophores, but the fluorescence of the fluorescent probe may be quenched which makes it difficult to observe and sensitive to environmental factors [77]. Compared to other fluorescence techniques, FP shows high-throughput screening and quick detection. Since this is an analytical method that relies on observing the changes of the global size of molecules, it can avoid the background fluorescence from other components in samples and the quenching of fluorophores cannot affect polarization (P) values. In addition, samples used for FP-based detection do not require pre-treatment, such as the separation step, which simplifies the sensing process to a considerable extent. In contrast with other kinds of fluorescent biosensors which are usually based on the changes of fluorescence intensity, FP values depend on the ratio of the maximum fluorescence intensity from the horizontal ($I_{⃫}$) and vertical ($I_{⊥}$) directions. This prevents the generation of fake signals caused by other factors, such as some proteins that can quench the light of fluorophores and sometimes the solvents that may also affect fluorescence intensity. Therefore, FP-based biosensors can be applied in many biological environments with high sensitivity to detect target analytes at low concentrations which allows FP-based biosensors to potentially be used in clinical settings for diagnostics.

Conventional FP-based biosensing systems use organic dyes as fluorophores to generate FP signals. However, organic dyes are sensitive to environmental factors such as pH and temperature. Compared to organic dyes, QDs with high photostability are excellent fluorescent labels; furthermore, QDs with surface functional groups can immobilize more receptors which can dramatically increase the sensitivity. Table 2 shows the FP biosensors made of QDs as compared with FP biosensors made of organic dyes.

According to previous studies, carbon-based nanomaterials, metal-based nanomaterials, semiconductor quantum dots, silicon-based nanoparticles, inorganic two-dimensional nanomaterials, and polystyrene nanoparticles have been used as amplifiers and involved in enhancing the sensitivity of FP-based biosensing systems. MWCNT as an amplifier has been used to detect DNA MTase. The FP value was amplified over 10 times compared to the system without MWCNTs. In addition, the dynamic range with MWCNTs can cover over 5 orders of magnitude from 10^−4^ U/mL to 10^2^ U/mL, while the range without MWCNTs is only from 1 U/mL to 10^2^ U/mL. Moreover, the limit of detection (LOD) of this biosensor is around 2 orders of magnitude lower than previously published approaches [24]. Ag_10_NPs used in a FP-based biosensing systems can produce a mass-augmented and metal-enhanced fluorescence (MEF) effect to enhance the FP signals which enhances the sensitivity of detection to around 3 orders of magnitude higher than published aptasensors with the LOD around 1.25 pM in the high-throughout detection of actual milk powder [32]. Gold nanoparticles using toehold-mediated strand-displacement reaction enhanced the FP-based detection of single nucleotide polymorphism, reaching a limit of detection at the nanomolar level. This strategy showed outstanding selectivity as 10 times the signal difference can be detected even if 100 times of the single-base mismatched strand were to exist [38]. A FP-based biosensor using PS NPs that assisted with the CSDA process had a limit of detection for the target protein in attomole range which is about 6 orders of magnitude lower than related sensors [76]. Meanwhile, inorganic two-dimensional nanosheets, such as MnO_2_ nanosheets, with their excellent ability to adsorb ssDNA, have also been investigated in related FP-based biosensing systems as enhancers. By using MnO_2_ nanosheets in this kind of biosensing system, the FP signal increased over 8-fold. In addition, a good linear relationship was obtained in a range of 30–240 nM for the detection of Ag^+^ [67].

Table 3 shows the advantages and disadvantages of nanomaterials used as amplifiers in FP biosensors in comparison with other types of amplifiers made of large biomolecules, such as proteins and nucleic acids.

The successful combination of FP technology and nanobiotechnology provides many new possibilities for using FP-based biosensors in sensitive and selective detections. Major benefits in commercialization of FP-based biosensing systems made of nanomaterials are as follows; (a) nanomaterials can minimize the size of the device; (b) the detection price could be reduced; (c) accuracy and sensitivity of the detection can be significantly enhanced; (d) lifetime of the product will be longer as nanomaterials have higher photostability.

## 5. Conclusions

In summary, fluorescence polarization technology can be used for determining low concentrations of analytes. In contrast with other fluorescent-based biosensing approaches which normally measure the fluorescence intensity or fluorescence wavelength, FP-based biosensors depend on the size of the fluorescent element. Therefore, the results of the biosensing process are not influenced by any quenching situations caused by the environment/solvents. Compared to conventional FP-based biosensors made of organic dyes, nanomaterials, with their high photostability, allow FP-based biosensors to detect different analytes with high sensitivity. In addition, nanomaterials used in FP-based biosensors can be applied in both generating and amplifying FP signals. In this review, different types of nanomaterials used in developing FP-based biosensors have been reviewed. The role of each type of nanomaterial acting as the fluorescent element and/or the signal amplifier has been discussed. Meanwhile, the advantages of the FP-based biosensing systems have been discussed and compared with other fluorescence-based techniques.

On the other hand, several challenges remain in employing nanomaterials for the development of FP-based biosensors. Firstly, though semiconductor QDs have been investigated in generating signals in FP biosensors, other types of nanomaterials, e.g., carbon QDs with tunable fluorescence, have not been addressed as the signal generator in FP-based biosensors. Secondly, fluorescence polarization is not only affected by the size of the fluorescent element, but also influenced by other factors, such as solvent viscosity, and the excited state lifetime of the fluorophore. However, most studies focus on mass-based approaches in amplifying FP signals. Thirdly, current FP biosensors focus on single-molecule detection. Nanomaterials, with their large surface area-to-volume ratio, can not only minimize the size of the biosensor, but can also possibly immobilize multiple receptors for multiplexed detection.

## Figures and Tables

**Figure 1 ijms-23-08625-f001:**
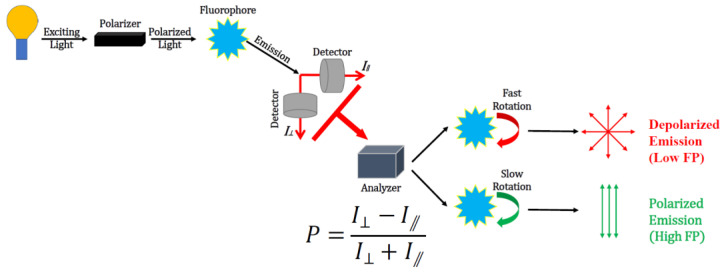
Schematic demonstration of the principle of fluorescence polarization.

**Figure 2 ijms-23-08625-f002:**
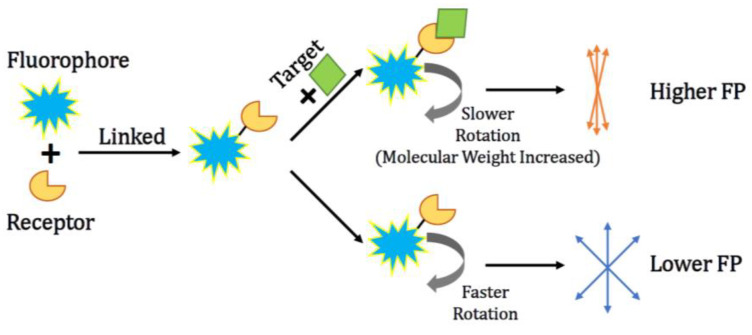
Mechanism of using fluorescence polarization for sensing.

**Figure 3 ijms-23-08625-f003:**
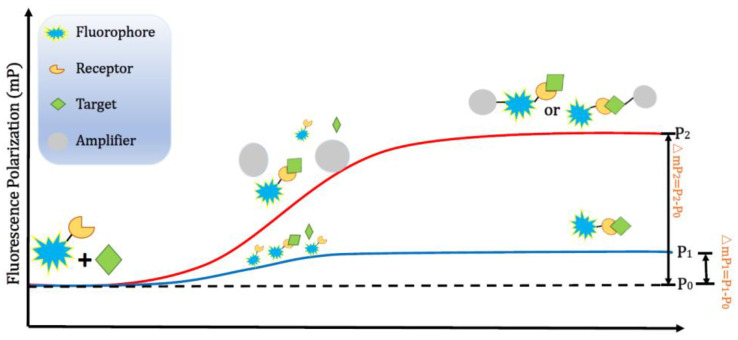
Strategy for amplifying the signal of fluorescence polarization.

**Figure 4 ijms-23-08625-f004:**
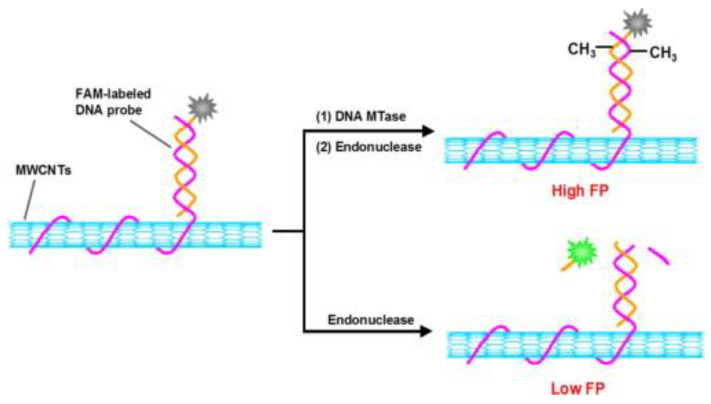
The principle of the MWCNT-based FP sensing platform used for the detection of DNA MTase. (Reprinted with permission from Ref. [24]. Copyright © 2014 Elsevier).

**Figure 5 ijms-23-08625-f005:**
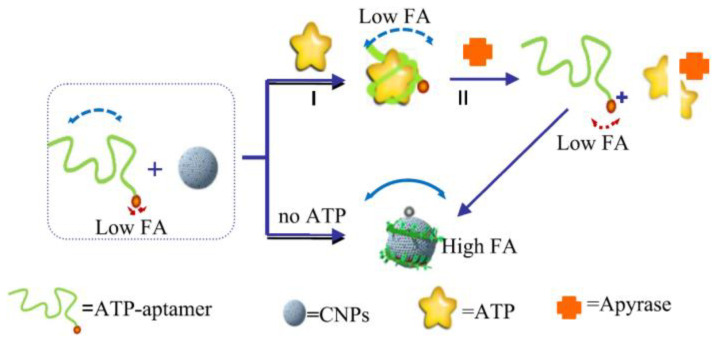
A schematic (not to scale) illustrating the CNP-based amplification fluorescent anisotropy detection apyrase. (Reprinted with permission from Ref. [26]. Copyright © 2014 Elsevier).

**Figure 6 ijms-23-08625-f006:**
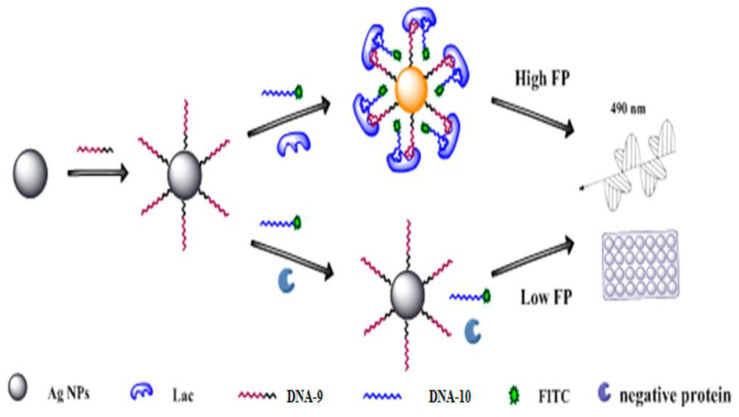
Principle of the Dual Amplified Aptasensor Based on Bivalent Aptamers and Ag_10_NPs Enhancement. Reprinted with permission from Ref. [32]. Copyright © 2017 American Chemical Society).

**Figure 7 ijms-23-08625-f007:**
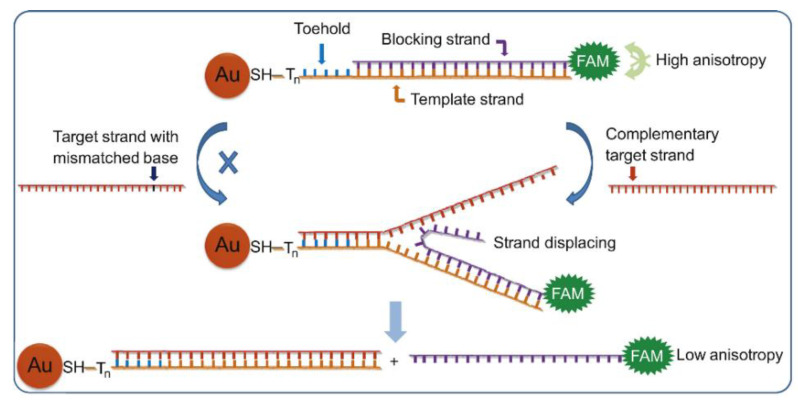
Schematic illustration of gold nanoparticle enhanced fluorescence polarization with the toehold-mediated strand-displacement reaction for single nucleotide polymorphisms detection. (Reprinted with permission from Ref. [38]. Copyright © 2013 Elsevier).

**Figure 8 ijms-23-08625-f008:**
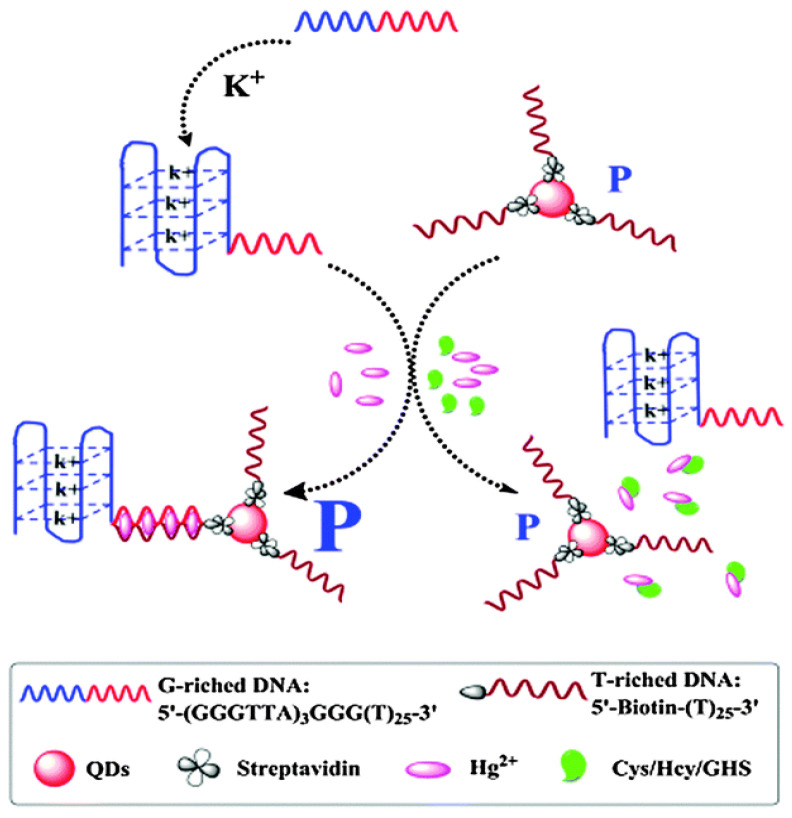
Schematic illustration of the QD fluorescence polarization enhancement homogenous system based on K^+^-mediated G-quadruplex formation for the determination of Hg^2+^ and biothiols. (Reprinted with permission from Ref. [45]. Copyright © 2014 ROYAL SOCIETY OF CHEMISTRY).

**Figure 9 ijms-23-08625-f009:**
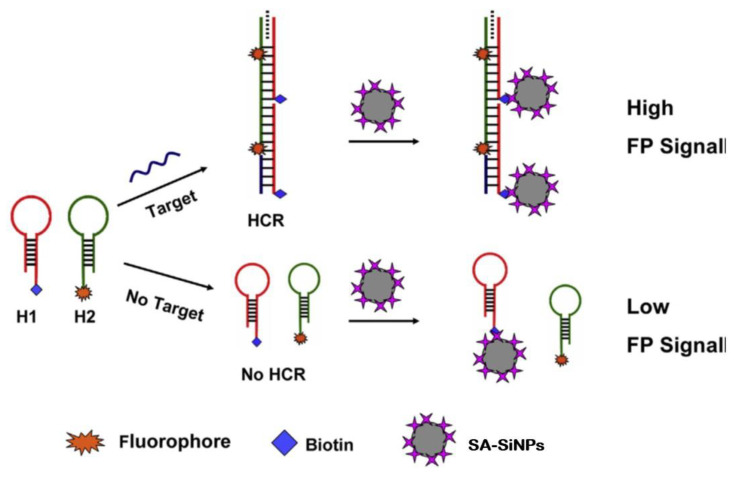
Schematic illustration of the amplified FP assay based on HCR and nanoparticles aggregation. (Reprinted with permission from Ref. [52]. Copyright © 2015 Elsevier).

**Figure 10 ijms-23-08625-f010:**
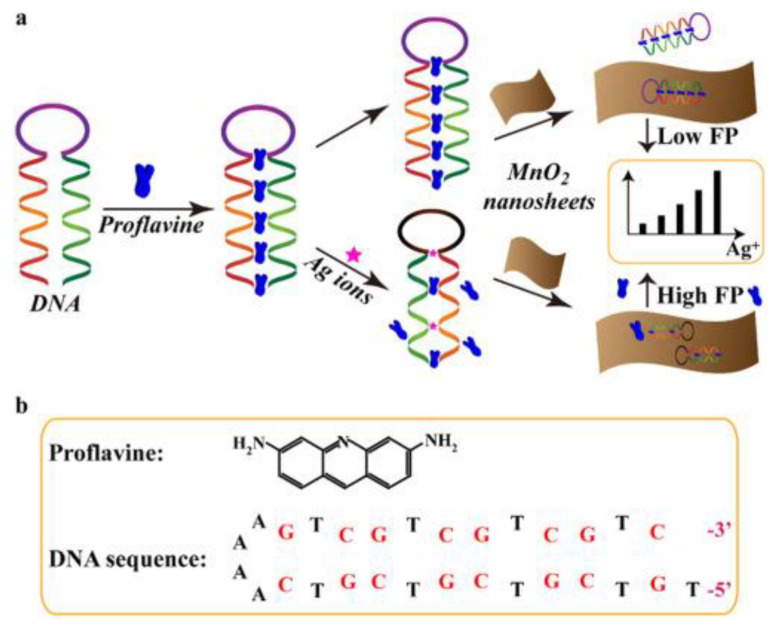
(**a**) Schematic illustration of Ag^+^-detecting MnO_2_ nanosheet-based fluorescence polarization biosensor. (**b**) Structure of proflavine and double-stranded DNA probe. (Reprinted with permission from Ref. [67]. Copyright © 2017 Elsevier).

**Figure 11 ijms-23-08625-f011:**
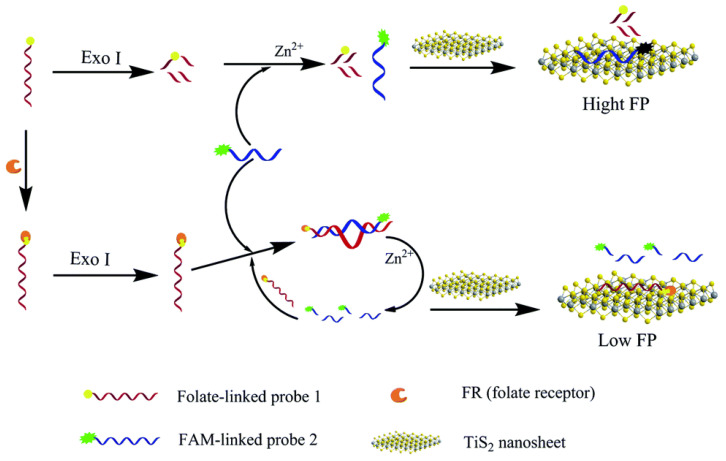
Schematic illustration of TiS_2_ nanosheet-enhanced fluorescence polarization biosensor relying on DNAzyme and small DNA probes for detecting FR. (Reprinted with permission from Ref. [69]. Copyright © 2016 Royal Society of Chemistry).

**Figure 12 ijms-23-08625-f012:**
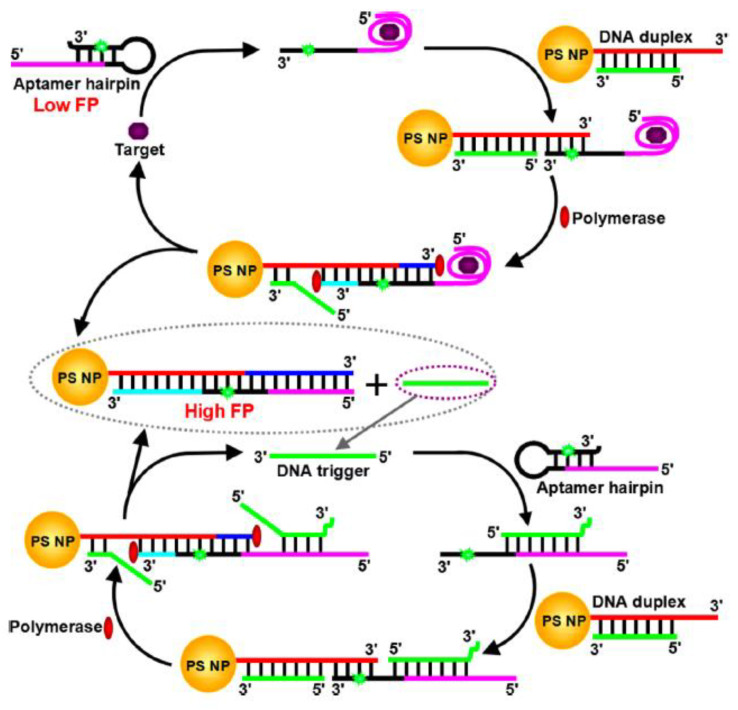
Principle of the Proposed Amplified FP Aptasensor Based on CSDA and PS NPs Enhancement. (Reprinted with permission from Ref. [72]. Copyright © 2015 American Chemical Society).

**Table 1 ijms-23-08625-t001:** Comparison of FP biosensors and other fluorescence sensing systems.

Method	Advantages	Disadvantages
Fluorescence Polarization (FP) -based biosensor	high selectivity and sensitivity [78]; independent of concentration [12]; no background fluorescence required [79]; no separation process [79]; signal amplifiable [80]; high-throughput [81]; real-time [26].	liquid-based sample. fluorophores may affect the binding ability of the receptor [82].
Fluorescence Resonance Energy Transfer (FRET) -based biosensor	high selectivity and sensitivity [83]; no separation process [84]; available in both liquid-base and solid-based.	efforts are required to choose suitable FRET pair [85].
Fluorescence Intensity (FI) Endpoint Measurement	simple steps for testing [86]; real-time data [87].	low signal-to-noise ratio [88]; susceptible to samples with self-fluorescence [89]; short lifetime of fluorophores [90].
Fluorescence Lifetime Imaging Microscopy (FLIM)	visualized [91]; probe molecular environments of fluorophores [92]; can distinguish fluorophore with overlap photoluminescence spectra.	instrument error [93]; skilled technician required [94]; the effectiveness of fluorescent probe may be affected by quenching [95]; easy to be affect by environmental factors.
Time-Resolved Fluoroimmunoassay (TRFIA)	low limit of detection [96]; a large range of linearity [97]; long lifetime of fluorophores [98].	expensive instrument [99]; environmental factors may affect the results.

**Table 2 ijms-23-08625-t002:** QDs vs. organic dyes in FP biosensors.

Materials	Advantages	Disadvantages
Quantum Dots (QDs)	full of functional groups on the surface for conjugation [100]; long fluorescence lifetimes [101]; tunable excitation wavelengths [102]; large surface area-to-volume ratio for immobilization [103]; multicolor experiments [104]; higher quantum yields [105]; higher thermal and photochemical stability [106].	Additional process required to make water soluble QDs. The toxicity of QDs could be an issue [107].
Other Dyes	easy to get [108]; small size [109]; good solubility [109].	high photobleaching rate [110]; sensitivity to pH changes [111]; lower thermal and photochemical stability [112]; expensive [112]; short fluorescence lifetimes [113].

**Table 3 ijms-23-08625-t003:** Comparison of using nanomaterials and other materials as amplifiers in FP-based biosensors.

Materials	Advantages	Disadvantages
Nanomaterials	easy to be modified [114]; minimize the size of sensing device [115]; size-dependent properties [116]; large surface area to volume ratio [117]; high photostability [117].	More studies are required to understand the impact of nanomaterials.
Other Amplifiers	good solubility [118].	signal quenching [119]; expensive process [120]; environment dependent [74].

## Data Availability

Not applicable.

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
