# Peer review of "Nanomaterials Used in Fluorescence Polarization Based Biosensors"

_ijms, 2022, doi:10.3390/ijms23158625_

Round 1

Reviewer 1 Report

General remark 1:

The whole paper needs to be subjected to editing of English language and style. Special attention has to be paid to repetitions, colloquialisms and imprecise statements. As an example:

-‘After Stokes first introduced “fluorescence” in 1852’

-Fluorescence analysis has demonstrated high sensitivity, low limit of detection, fast and high-throughput, and its applications have been involved in sensing, drug delivery and in vivo imaging.’

-‘In the presence of HIV-DNA, AuNPs-DNA dendritic macromolecules and fluorescently labeled DNA probes bound to HIV-DNA in a sandwich format to form a conjugate’

-‘He and colleagues developed a fairly sensitive and selective method (…)’

-‘In this decade, CdTe QDs have also been tried in FP-based biosensors for generating FP signals.’

-‘no background fluorescence requires’ (table 1)

-‘lanthanide ions are unfriendly to the environment’

General remark 2:

Manuscript lacks references to back-up following statements:

-‘The value of FA has a relationship with that of FP which can be defined as equation (3)’ (lines 61-63)

-‘In comparison to other biomolecules used to amplify the signal, nano-materials that have higher stability and larger volume/mass, such as graphene oxide and gold nanoparticles, show great potential as amplifiers to enhance the performance of sensing systems.’ (lines 99-102)

-‘In addition, nanomaterials are rich in multiple functional groups on the surface, can easily be modified and can immobilize receptors.’ (lines 110-112)

-‘Therefore, various nanomaterials have been investigated as amplifiers in mass-based approaches to enhance the FP signals for optimizing the sensitivity of FP-based assays’ (lines 117-118)

-‘Because of their unique optoelectronic properties and excellent biocompatibility, AuNPs have attracted extensive interest in the field of bioanalysis.’ (lines 252-253)

-‘AuNPs can easily combine with biomolecules such as proteins and DNA via self-assembly and maintain the biochemical activity of the labeled biomolecules’ (lines 255-257)

-‘NIR light, defined as light with wavelengths above 650 nm (…)’ (line 420)

Remark 3:

‘Normally, a fluorophore labeled with a receptor (fluorescent-receptor) will display a smaller P value as the small molecule exhibits fast rotation diffusion in solution’ (lines 75-76)

a)Smaller P value than what?

b)Term ‘small molecule’ in pharmacology/molecular biology is dedicated to molecules <1000 Da. It is in contrary to ‘fluorophore labeled with a receptor’.

c)Usually fluorophores are used to label receptors instead of the opposite (labeling fluorophores with receptors).

Remark 4:

‘When the target exists, the probe specifically bonds to the target making the weight of global probe increase dramatically which decreases the rotation rate, resulting in a higher P value’ (lines 82-84)

Significant part of publications concerning fluorescence polarization assays cover small molecule (i.e. <1000 Da) screening. In such case the sentence above is not applicable. More universal example would be beneficial for future reader.

Remark 5:

‘Carbon-based materials have recently become valuable’ (lines 126-127)

Sentence suggest that carbon-based materials were not valuable in the past.

Remark 6:

‘It has been found that single-stranded DNA (ssDNA) is strongly adsorbed on CNTs by π–π interactions and van der Waals interactions, while double-stranded DNA (dsDNA) can helically wrap around CNTs upon thermal denaturation to unzip the two strands.’ (lines 140-143)

Sentence should be rewritten to clearly state the nature of interaction between dsDNA and CNTs. 

Remark 7:

‘These different interactions on CNTs caused by DNA hybridization could be used in developing DNA sensors ‘ (lines 143-144)

Sentence should be rewritten to clearly state that it is not the hybridization phenomenon that caused interactions CNT-DNA, but the properties of ssDNA and dsDNA.

Remark 9:

‘Huang and co-workers also used the same principle with MWCNTs to detect protease. The detection limits of this assay for thrombin and chymotrypsin were 0.5 pM and 0.3 pM, respectively.’ (lines 162-164)

The study should be described in more detailed manner, as it is done for the other references.

Remark 10:

Hydrolyze nucleoside triphosphates, an integral membrane protein, could hydrolyze ATP and disassemble the P-ATP complex, which would lead to an increase in FP.’ (lines 170-172)

Improper name of enzyme was used.

Remark 11:

Lines 177-181; Reference [28]

The reference does not describe application of fluorescence polarization in assay.

Remark 12:

Reference [33]

Reference is not written properly.

Remark 13:

Lines 223-234 (Reference [34])

Paragraph should be rewritten, to clearly present the study goal, methodology and conclusions.

Remark 14:

‘Their results indicated that this amplification strategy was the most effective method for ultrasensitive determination of oligonucleotides using fluorescence polarization.’ (lines 274-277)

No information on other methods evaluated by Liang et al. is provided in the manuscript.

Remark 15:

‘the linear relationship of ATP concentration ranged from 8 × 10^-12 M to 2.40 × 10^-4 M’ (lines 285-286)

‘Firstly, it was highly sensitive with a detection limit of 0.2 ppb’ (lines 315-316)

Approach for presentation of concentration should be unified along the manuscript.

Remark 16:

‘The detection sensitivity could be significantly improved to 1.0 nM using a “gold nanoparticle enhancement” method.’ (lines 309-311)

‘Firstly, it was highly sensitive with a detection limit of 0.2 ppb’ (lines 315-316)

A method's sensitivity is a measure of its ability to establish that such a difference is significant. Sensitivity is often confused with a method's detection limit, which is the smallest amount of analyte we can determine with confidence.

Remark 17:

‘In addition, samples used for FP-based detection do not require pre-treatment, such as the separation step, which simplifies the sensing process to a large extent.’ (lines 504-506)

Addition of reagents (i.e. fluorophore-labeled receptors) should be considered as a pre-treatment.

Remark 18:

Discussion + Conclusions (lines 480-541)

Authors should expand sections by discussing the:

-Future of the Fluorescence Polarization-based biosensors

-Currently commercialized biosensors employing such combination (nanomaterials + FP), or potential of commercialization if there are none. 

Additional questions to Authors:

Question 1: 

Can assay done in laboratory-based Microplate Reader be regarded as biosensor? What is the difference between an such assay and a biosensor?

Question 2: 

What was the methodology for searching and selecting the research articles presented in this manuscript?

Author Response

Dear Editors and Reviewers,

Thank you for giving us an opportunity to revise our manuscript (ijms-1815998), entitled " Nanomaterials Used in Fluorescence Polarization Based Biosensors". We greatly appreciate you and reviewers for your time and constructive comments. We had carefully considered and taken all the reviewers’ comments and revised the manuscript. It is our belief that the manuscript is substantially improved after making the suggested edits. The authors welcome further constructive comments if any. Below is our point-by-point response to the comments of reviewers.

General remark 1:

The whole paper needs to be subjected to editing of English language and style. Special attention has to be paid to repetitions, colloquialisms and imprecise statements. As an example:

-‘After Stokes first introduced “fluorescence” in 1852’

-Fluorescence analysis has demonstrated high sensitivity, low limit of detection, fast and high-throughput, and its applications have been involved in sensing, drug delivery and in vivo imaging.’

-‘In the presence of HIV-DNA, AuNPs-DNA dendritic macromolecules and fluorescently labeled DNA probes bound to HIV-DNA in a sandwich format to form a conjugate’

-‘He and colleagues developed a fairly sensitive and selective method (…)’

-‘In this decade, CdTe QDs have also been tried in FP-based biosensors for generating FP signals.’

-‘no background fluorescence requires’ (table 1)

-‘lanthanide ions are unfriendly to the environment’

Authors response:

We appreciate for the reviewer’s time and comments. The manuscript has been revised and edited. These sentences have been rewritten as well.

General remark 2:

Manuscript lacks references to back-up following statements:

-‘The value of FA has a relationship with that of FP which can be defined as equation (3)’ (lines 61-63)

-‘In comparison to other biomolecules used to amplify the signal, nano-materials that have higher stability and larger volume/mass, such as graphene oxide and gold nanoparticles, show great potential as amplifiers to enhance the performance of sensing systems.’ (lines 99-102)

-‘In addition, nanomaterials are rich in multiple functional groups on the surface, can easily be modified and can immobilize receptors.’ (lines 110-112)

-‘Therefore, various nanomaterials have been investigated as amplifiers in mass-based approaches to enhance the FP signals for optimizing the sensitivity of FP-based assays’ (lines 117-118)

-‘Because of their unique optoelectronic properties and excellent biocompatibility, AuNPs have attracted extensive interest in the field of bioanalysis.’ (lines 252-253)

-‘AuNPs can easily combine with biomolecules such as proteins and DNA via self-assembly and maintain the biochemical activity of the labeled biomolecules’ (lines 255-257)

-‘NIR light, defined as light with wavelengths above 650 nm (…)’ (line 420)

Authors response:

Authors are thankful for the comments. Additional references have been included to support the statements.

Remark 3:

‘Normally, a fluorophore labeled with a receptor (fluorescent-receptor) will display a smaller P value as the small molecule exhibits fast rotation diffusion in solution’ (lines 75-76)

  1. Smaller P value than what?

Authors response: It refers a lower degree of fluorescence polarization. The sentence is corrected.

  1. Term ‘small molecule’ in pharmacology/molecular biology is dedicated to molecules <1000 Da. It is in contrary to ‘fluorophore labeled with a receptor’.

Authors response: We apologize the unclear statement. Correct statement is included in the revised manuscript. We actually meant that fluorophore-labeled receptor has smaller size/molecule weight than fluorophore-labeled receptor bound to the analyte.

  1. Usually fluorophores are used to label receptors instead of the opposite (labeling fluorophores with receptors).

Authors response: We apologize the unclear statement. Correct statement is included in the revised manuscript. Remark 4:

‘When the target exists, the probe specifically bonds to the target making the weight of global probe increase dramatically which decreases the rotation rate, resulting in a higher P value’ (lines 82-84)

Significant part of publications concerning fluorescence polarization assays cover small molecule (i.e. <1000 Da) screening. In such case the sentence above is not applicable. More universal example would be beneficial for future reader.

Authors response: We apologize the unclear statement. Correct statement is included in the revised manuscript.

Remark 5:

‘Carbon-based materials have recently become valuable’ (lines 126-127)

Sentence suggest that carbon-based materials were not valuable in the past.

Authors response: The sentence is revised.

Remark 6:

‘It has been found that single-stranded DNA (ssDNA) is strongly adsorbed on CNTs by π–π interactions and van der Waals interactions, while double-stranded DNA (dsDNA) can helically wrap around CNTs upon thermal denaturation to unzip the two strands.’ (lines 140-143)

Sentence should be rewritten to clearly state the nature of interaction between dsDNA and CNTs.

Authors response: the sentences are revised.

Remark 7:

‘These different interactions on CNTs caused by DNA hybridization could be used in developing DNA sensors ‘ (lines 143-144)

Sentence should be rewritten to clearly state that it is not the hybridization phenomenon that caused interactions CNT-DNA, but the properties of ssDNA and dsDNA.

Authors response: CNTs can be used as amplifier to enhance the FP signal. The sentences are revised.

Remark 9:

‘Huang and co-workers also used the same principle with MWCNTs to detect protease. The detection limits of this assay for thrombin and chymotrypsin were 0.5 pM and 0.3 pM, respectively.’ (lines 162-164)

The study should be described in more detailed manner, as it is done for the other references.

Authors response: The manuscript has been revised; more in-depth description of the study is included.

Remark 10:

Hydrolyze nucleoside triphosphates, an integral membrane protein, could hydrolyze ATP and disassemble the P-ATP complex, which would lead to an increase in FP.’ (lines 170-172)

Improper name of enzyme was used.

Authors response: We are thankful for the reviewer’s time and comment. It has been revised.

Remark 11:

Lines 177-181; Reference [28]

The reference does not describe application of fluorescence polarization in assay.

Authors response: We sincerely apologize. We double-checked all the other references.

Remark 12:

Reference [33]

Reference is not written properly.

Authors response: The reference number has been revised as [40].

Remark 13:

Lines 223-234 (Reference [34])

Paragraph should be rewritten, to clearly present the study goal, methodology and conclusions.

Authors response: the paragraph was rewritten.

Remark 14:

‘Their results indicated that this amplification strategy was the most effective method for ultrasensitive determination of oligonucleotides using fluorescence polarization.’ (lines 274-277)

No information on other methods evaluated by Liang et al. is provided in the manuscript.

Authors response: the paragraph was rewritten.

Remark 15:

‘the linear relationship of ATP concentration ranged from 8 × 10^-12 M to 2.40 × 10^-4 M’ (lines 285-286)

‘Firstly, it was highly sensitive with a detection limit of 0.2 ppb’ (lines 315-316)

Approach for presentation of concentration should be unified along the manuscript.

Authors response: Here, 0.2 ppb is equal to 1.0 nM, and the sentence has been rewritten.

Remark 16:

‘The detection sensitivity could be significantly improved to 1.0 nM using a “gold nanoparticle enhancement” method.’ (lines 309-311)

‘Firstly, it was highly sensitive with a detection limit of 0.2 ppb’ (lines 315-316)

A method's sensitivity is a measure of its ability to establish that such a difference is significant. Sensitivity is often confused with a method's detection limit, which is the smallest amount of analyte we can determine with confidence.

Authors response: Authors are thankful for the reviewer’s commentThe manuscript has been revised to indicate the difference between the method’s sensitivity and detection of limit.

Remark 17:

‘In addition, samples used for FP-based detection do not require pre-treatment, such as the separation step, which simplifies the sensing process to a large extent.’ (lines 504-506)

Addition of reagents (i.e. fluorophore-labeled receptors) should be considered as a pre-treatment.

Authors response: The addition of reagents is not considered as a typical pre-treatment process in detection. Pre-treatment procedures usually include processes such as filtration & centrifugation, solvent extraction, supercritical fluid extraction, solid-phase extraction, solid-phase microextraction, liquid-phase microextraction or derivatization. Furthermore, biosensors usually do not require such pre-treatment processes. Biosensors are usually developed by linking the bioreceptor and the transducer together acting as a probe where the bioreceptor has high affinity to a specific target analyte.

Remark 18:

Discussion + Conclusions (lines 480-541)

Authors should expand sections by discussing the:

-Future of the Fluorescence Polarization-based biosensors

-Currently commercialized biosensors employing such combination (nanomaterials + FP), or potential of commercialization if there are none.

Authors response: The sections of Discussions and Conclusions have been rewritten. 

Additional questions to Authors:

Question 1:

Can assay done in laboratory-based Microplate Reader be regarded as biosensor? What is the difference between an such assay and a biosensor?

Authors response: Yes, bioassay is normally considered as a solution-based biosensors.[1] Biosensor is an analytical device and is a short form of “biological sensor”. Similar to other kinds of sensors, such as physical or chemical sensors, all sensors are used for a kind of detection and translate a process, which is hard to determine, into a kind of readable signal. Unlike other kinds of sensors, biosensors use bio-elements as a receptor to recognize what happened in the sensing system.

Basically, biosensors can be separated into three main parts: the recognizing part, transducing part and the processing part. Usually, bioreceptors are linked with transducers to recognize targets or biological events followed by transducing optical, electrical, thermal, or other types of signals for analysis. If all three of these conditions are satisfied, it can be called biosensor.

The mechanisms of assays developed for detection are also used in developing biosensing systems and these assays can be easily carried out in multi-well plates which can obtain high throughput applications. Basically, each well is a biosensor.

Similarly, ELISA (enzyme-linked immunosorbent assay) can also act as a biosensor which is mainly based on antigen/antibody interactions. By using ELISA kits to measure the concentration of target analytes, the physiochemical change is due to the conversion of substrate catalyzed by the enzyme to colored products which can be read by a microplate reader. And for FP-based assays, a microplate reader is used to analyze the changes of FP.

Reference:

[1] Ugo, Paolo, Marafini, Pietro and Meneghello, Marta. "2 Introduction to bioanalytical assays and biosensors". Bioanalytical Chemistry: From Biomolecular Recognition to Nanobiosensing, Berlin, Boston: De Gruyter, 2021, pp. 61-102. https://doi.org/10.1515/9783110589160-002

Question 2:

What was the methodology for searching and selecting the research articles presented in this manuscript?

Authors response: Both “google scholar” and “Western digital library” are used in searching the literatures related to the topic of nanomaterials used in FP-based bisoensors. The key words are as follows “nanomaterial/nanoparticle”, “fluorescence polarization” and “biosensor”. In addition, we rechecked by searching “name of NPs + fluorescence polarization + biosensor” and “name of NPs + fluorescence anisotropy + biosensor” to ensure we did not miss any articles. Based on all related literatures, nanomaterials used in FP-based biosensors are classified into eight categories.  

Reviewer 2 Report

The authors reported a review on Nanomaterials Used in Fluorescence Polarization Biosensors as a signal amplifier. The use of nanoparticles to amplify the detection signal has been reported widely in recent years. Totally speaking, the methods reported in this paper lacks of novelty to be fair this can’t be accepted as a review. Lacks examples and at the same time in the recent literature.

I recommend the manuscript for publishing after the significant revision indicated below.

Specific comments:

  1. In my opinion, the manuscript presented for review does not contain any novelty. If you google the keywords, you will find some reviews that are not cited in your manuscript and related to your subject.
  2. Also, the use of nanomaterials as an analytical signal amplifier is not new (the first examples of a review after entering the keyword in the web browser (like Sensors 2020, 20, 7132; doi:10.3390/s20247132; RSC Adv., 2022, 12, 6364-6376; Analytical and Bioanalytical Chemistry Volume 412, pages6655–6665 (2020); …)
  3. Please highlight the novelty element in the review with the updates in the field more precisely and with examples and illustrations. If the Authors believe that this work has new elements, I would like to ask you to emphasize them.
  4. Very messy drawing descriptions. It must be corrected

Author Response

Dear Editors and Reviewers,

Thank you for giving us an opportunity to revise our manuscript (ijms-1815998), entitled " Nanomaterials Used in Fluorescence Polarization Based Biosensors". We greatly appreciate you and reviewers for your time and constructive comments. We had carefully considered and taken all the reviewers’ comments and revised the manuscript. It is our belief that the manuscript is substantially improved after making the suggested edits. The authors welcome further constructive comments if any. Below is our point-by-point response to the comments of reviewers.

  1. In my opinion, the manuscript presented for review does not contain any novelty. If you google the keywords, you will find some reviews that are not cited in your manuscript and related to your subject.

Authors response: Authors appreciate the review’s time. Fluorescence polarization has been used in different areas. Conventional FP-based biosensors are made of organic dyes. Recently, different types of nanomaterials have been applied in FP-based biosensors; while very few review papers have been reported on using nanomaterials for FP-based biosensors. Herein, two main motivations are: firstly, we would like to introduce FP technology to researchers in the field of detection chemicals/biomolecules. Secondly, this review paper is the first one provide eight categories of nanomaterials used in FP-based biosensors that can not only be employed for generating fluorescent signals but can also be considered as signal amplifiers for enhancing the sensitivity through increasing the fluorescent polarization.

In addition, our topic is to review studies combine the “FP technology” and “nanobiotechnology”. So, references we reviewed here must be related to both fields and we have almost reviewed all suitable articles and classified them under different sections. Publications which did not use nanomaterials in developing FP-based biosensors or in developing other kinds of biosensors were not included.

2. Also, the use of nanomaterials as an analytical signal amplifier is not new (the first examples of a review after entering the keyword in the web browser (like Sensors 2020, 20, 7132; doi:10.3390/s20247132; RSC Adv., 2022, 12, 6364-6376; Analytical and Bioanalytical Chemistry Volume 412, pages6655–6665 (2020); …)

Authors response: It is noted that some publications on reviewing the applications of FP technology. However, we are focusing on different scope with different motivations.

For the “Hendrickson, O. D., Taranova, N. A., Zherdev, A. V., Dzantiev, B. B., & Eremin, S. A. (2020). Fluorescence polarization-based bioassays: new horizons. Sensors, 20(24), 7132.”:

Firstly, they were focusing on reviewing the possibility of using FP-based bioassays in different kinds of detections. But the major purpose of our review is to demonstrate the possibility of using nanomaterials in developing FP-based biosensors. Secondly, they only compared the FPIA with other kind of approaches of detection, such as ELISA and HPLC. However, we gave the comparison among using FP-based biosensors and other kinds of fluorescent-based technologies, such as FRET, which is more specific to our topic. Meanwhile, we also compared using nanomaterials and other kinds of materials in developing FP-based biosensors.

For the “Xiao, X., & Zhen, S. (2022). Recent advances in fluorescence anisotropy/polarization signal amplification. RSC advances, 12(11), 6364-6376.”

As can be seen from the title, they were focused on reviewing the strategies on amplifying FP signals. However, nanomaterials can be not only used in amplifying FP signals but can also be able to generate FP signals, such as CdTe quantum dots. Moreover, in our study, we reviewed nanomaterials to be used in both generating and amplifying FP signals. Secondly, our review is discussing the biosensors, but the above reference has a different focus, rather than the biosensing systems.

  1. Please highlight the novelty element in the review with the updates in the field more precisely and with examples and illustrations. If the Authors believe that this work has new elements, I would like to ask you to emphasize them.

Authors response: Firstly, very few publications focused on using nanomaterials in developing FP-based biosensors. Secondly, few published paper reviewed different categories of nanomaterials in FP-based biosensors that can not only be employed for generating fluorescent signals but can also be considered as signal amplifiers for enhancing the sensitivity through increasing the fluorescent polarization.  In addition, this manuscript displaces the advantages of FP-based biosensors made of nanomaterials as compared with conventional FP-based biosensors made of organic molecules. We think that this review provides a comprehensive insight on employing nanomaterials in developing FP-based biosensors.

4. Very messy drawing descriptions. It must be corrected.

Authors response: Authors appreciate the review’s time. The manuscript has been revised and edited.

Round 2

Reviewer 1 Report

Authors did improve the manuscript significantly.

Nevertheless:

1) it should undergo additional language check, as there are still some minor grammar/ syntax errors.

2) Authors need to check if every abbreviation is defined properly.

3) Huang et al. group (line 250) is the same group as described earlier (line 219)

Author Response

Dear Editor and Reviewers,

Thank you for giving us an opportunity to revise our manuscript (ijms-1815998), entitled " Nanomaterials Used in Fluorescence Polarization Based Biosensors". We greatly appreciate you and reviewers for your time and constructive comments. We had carefully considered and taken all the reviewers’ comments and revised the manuscript. Below is our point-by-point response to the comments of reviewers (in blue).

Authors did improve the manuscript significantly.

Nevertheless:

  • it should undergo additional language check, as there are still some minor grammar/ syntax errors.

Authors response: Authors are thankful for the reviewer’s time. The manuscript has been revised and edited as per the comments of all reviewers. Proofreading was finished by a native English-speaking colleague as well.

  • Authors need to check if every abbreviation is defined properly.

Authors response: All abbreviations have been defined.

  • Huang et al. group (line 250) is the same group as described earlier (line 219)

Authors response: Authors apologize for this error. “Huang et al. group” in previous version of the manuscript, in Line 153 and Line 160 refers the same group. A correction is made (please see the revised manuscript, Line 154).

Reviewer 2 Report

The authors reported a review on Nanomaterials Used in Fluorescence Polarization Biosensors as a signal amplifier. I recommend the manuscript for publishing after the minor revision indicated below.

Specific comments:

1. I will suggest adding some additional images in different sections to the authors.

3.1.2 Carbon nanoparticles

3.2.1. Silver Nanoparticles

3.4 Silica Nanoparticles

3.7. Polystyrene nanoparticle

Author Response

Dear Editors and Reviewers,

Thank you for giving us an opportunity to revise our manuscript (ijms-1815998), entitled " Nanomaterials Used in Fluorescence Polarization Based Biosensors". We greatly appreciate you and reviewers for your time and constructive comments. We had carefully considered and taken all the reviewers’ comments and revised the manuscript. Below is our point-by-point response to the comments of reviewers (in blue).

The authors reported a review on Nanomaterials Used in Fluorescence Polarization Biosensors as a signal amplifier. I recommend the manuscript for publishing after the minor revision indicated below.

Specific comments:

  1. I will suggest adding some additional images in different sections to the authors.

3.1.2 Carbon nanoparticles

Authors response: Figure 5 is included to indicate carbon nanoparticles used in FP biosensor.

3.2.1. Silver Nanoparticles

Authors response: Figure 6 is included to indicate carbon nanoparticles used in FP biosensor.

3.4 Silica Nanoparticles

Authors response: Figure 9 is included to indicate carbon nanoparticles used in FP biosensor.

3.7. Polystyrene nanoparticle

Authors response: Figure 12 is included to indicate carbon nanoparticles used in FP biosensor.